# Exploring the Dynamic Spatio-Temporal Correlations between PM_2.5_ Emissions from Different Sources and Urban Expansion in Beijing-Tianjin-Hebei Region

**DOI:** 10.3390/ijerph18020608

**Published:** 2021-01-12

**Authors:** Shen Zhao, Yong Xu

**Affiliations:** 1Institute of Geographic Sciences and Natural Resources Research, Chinese Academy of Sciences, Beijing 100101, China; zhaos.14s@igsnrr.ac.cn; 2College of Resources and Environment, University of Chinese Academy of Sciences, Beijing 100049, China

**Keywords:** PM_2.5_ emission, urban expansion, spatio-temporal analysis, Beijing-Tianjin-Hebei region

## Abstract

Due to rapid urbanization globally more people live in urban areas and, simultaneously, more people are exposed to the threat of environmental pollution. Taking PM_2.5_ emission data as the intermediate link to explore the correlation between corresponding sectors behind various PM_2.5_ emission sources and urban expansion in the process of urbanization, and formulating effective policies, have become major issues. In this paper, based on long temporal coverage and high-quality nighttime light data seen from the top of the atmosphere and recently compiled PM_2.5_ emissions data from different sources (transportation, residential and commercial, industry, energy production, deforestation and wildfire, and agriculture), we built an advanced Bayesian spatio-temporal autoregressive model and a local regression model to quantitatively analyze the correlation between PM_2.5_ emissions from different sources and urban expansion in the Beijing-Tianjin-Hebei region. Our results suggest that the overall urban expansion in the study area maintained gradual growth from 1995 to 2014, with the fastest growth rate during 2005 to 2010; the urban expansion maintained a significant positive correlation with PM_2.5_ emissions from transportation, energy production, and industry; different anti-haze policies should be designated according to respective local conditions in Beijing, Tianjin, and Hebei provinces; and during the period of rapid urban expansion (2005–2010), the spatial correlations between PM_2.5_ emissions from different sources and urban expansion also changed, with the biggest change coming from the PM_2.5_ emissions from the transport sector.

## 1. Introduction

China’s urbanization has resulted in significant achievements in recent decades, with an increasing number and continuously expanding scale of cities [1,2]. The level of urbanization has increased from 17.92% in 1978 to 60.60% in 2019 [3]. However, behind these achievements, China has also become the country with the largest pollutant emissions in the world, resulting in various environmental pollution problems, particularly the deterioration of air quality, which is serious in terms of its impact range and intensity [4,5]. The main reason for the deterioration of air quality in urban areas is pollutant emissions, such as SO_2_, NO_x_, and PM_2.5_ (particulate matters with an aerodynamic diameter ≤ 2.5 μm) [6,7]. Existing research has found that PM_2.5_ emission is the main pollutant in haze pollution in China and presents typical regional characteristics [8,9]. Due to its long residence time in the atmosphere, it can not only reduce the visibility of the atmosphere and lead to climate change, but can also affect human health [10,11,12]. Medical research has proved that PM_2.5_ pollutants can cause various respiratory diseases and increase the death rate of exposed people by destroying the human immune system [13,14].

The Beijing-Tianjin-Hebei region is one of the most important urban agglomerations in China. The extremely high population density and limited urban living space have led to a series of problems that need to be resolved, including haze pollution and the resulting health problems [15,16]. In 2013, the air quality qualified days in the Beijing-Tianjin-Hebei region accounted for only 37.5% of the whole year, and seven of the ten cities with the worst air quality in the country were in the Beijing-Tianjin-Hebei region. In 2014, haze pollution occurred in most provinces in central and eastern China, with a heavy haze area of approximately 8.1 × 105 km^2^, mainly concentrated in the Beijing-Tianjin-Hebei region. In particular, according to the research results of Song et al. [17], in 2015, about half of the population living in Beijing-Tianjin-Hebei Region was exposed to areas where the average monthly PM_2.5_ concentration was higher than 80 μg/m^3^, and in terms of all-cause diseases, cardiovascular diseases, and respiratory diseases, the estimated number of premature deaths attributed to PM_2.5_ pollution were 138,150, 80,945 and 18,752, respectively.

A large number of empirical studies have demonstrated the relationships between pollutant emissions, urbanization, and economic development [18,19,20,21,22]. However, these studies rarely focus on the sources of PM_2.5_ emissions to explain the differences in correlations between these variables and urban expansion. Moreover, few studies have considered both temporal dynamics and spatial correlations simultaneously when modeling pollutant emissions data and urban expansion data [23,24]. Furthermore, most of these studies rely on large-scale analysis units (provincial units or municipal units) [25,26], which may lead to the potential for heterogeneity within the unit to be ignored. In addition, the time span of the selected environmental pollution data is usually short (for instance, 5 to 10 years) [27,28], which may reduce the reliability and accuracy of the correlation between urban expansion and pollutant emissions estimated by the model. To address the above problems, there is an urgent need for high-quality data sets of various sources and suitable models that can handle the complex spatio-temporal effects existing in the data sets. The purpose of this paper is not to prove that there is a simple one-way causality between urban expansion and PM_2.5_ emission. In this paper, the pollutant emission data source used was PM_2.5_ emission data from different sectors, so the PM_2.5_ emission data used is more like an intermediate variable connecting urban expansion and various sectors (such as transportation, energy production, and industry). When studying the correlation between urban expansion and various sectors in the process of urbanization, it is difficult to find an indicator that is convenient for unified quantification and can simultaneously characterize the development degree of different sectors. PM_2.5_ emissions from different sources can serve as such an indicator to effectively solve this problem, which is a manifestation of the innovation of this study.

Therefore, this article attempts to solve these problems and provide reliable and specific estimates of the correlation between PM_2.5_ emission and urban expansion. First, we adopted PM_2.5_ emission data that has good spatial resolution and a long-time span (20 years), and has been classified according to the sources of emissions, which were tested for validity in previous studies. Then, we used remote sensing data to characterize urban expansion, which went beyond the traditional method of using statistical data to measure urban expansion. Finally, this study established a newly developed advanced Bayesian spatio-temporal autoregressive model and a local regression model to estimate the spatio-temporal dynamic effects between PM_2.5_ pollution and urbanization through accurate and flexible modeling.

The remainder of this paper is structured as follows. The second section introduces the research area, data and its processing, research methods, and related models. Section 3 describes the findings of the descriptive analysis and spatio-temporal models. In Section 4, we discuss possibilities based on simulation results. Finally, Section 5 summarizes these research findings.

## 2. Materials and Methods

### 2.1. Overview of the Study Area

The study area is composed of 204 county-level administrative units (16 counties in Beijing, 16 counties in Tianjin, and 172 counties in Hebei Province) throughout the Beijing-Tianjin-Hebei region, located in north China (36°05′–42°40′ N, 113°27′–119°50′ E) (Figure 1). The Beijing-Tianjin-Hebei region is China’s “capital economic circle” and one of the most densely populated urban areas in China. In 2019, the land area (1.24 × 105 km^2^) of the Beijing-Tianjin-Hebei region accounted for only 2.3% of the country, but concentrated 8.1% of the country’s population (1.13 × 10^8^) and 9.7% of the county’s GDP (8.46 × 10^12^ Yuan). The urbanization rate is 86.60% in Beijing, 83.48% in Tianjin, and 57.60% in Hebei Province [3].

### 2.2. Remote Sensing Data

In this study, we used nighttime light data to characterize urban expansion. Nighttime light data has been increasingly used as an indicator of urban expansion and socio-economic development [29,30,31,32,33]. The source of nighttime light data for this paper was NOAA/NGDC (https://www.ngdc.noaa.gov/ngdc.html), with a fine spatial resolution of 1 × 1 km in the matched time periods. Data was de-clouded, and background noise and short-time data (volcanic gas, forest fire, aurora, etc.) were eliminated. The gray value of the pixels was between 0 and 63. We first extracted the nighttime light data for Beijing-Tianjin-Hebei region from 1995 to 2014, then we divided the total luminosity of each county by its total area to characterize each county’s urban expansion. Here we used luminosity density (LD) to represent this variable (Table 1). The standard GIS areal weighting approach was involved during the LD data process.

The standard GIS areal weighting approach used in this paper was based on the method of Lloyd et al. [34]. Areal weighting entails the overlay of the source zones (nighttime light data or pollutant emissions data), *s*, and the target zones, *t* (1 km grid cells), and the proportional allocation of raw data to the target zones which it overlays:(1)$$Z_{t}=\overset{N}{\underset{s=1}{\sum }}\frac{A_{st}}{A_{s}}Z_{s}$$
where $Z_{t}$ is the estimated value (nighttime light data or pollutant emissions data) of the target zone t, $A_{st}$ is the area of the zone of intersection between s and t and $A_{s}$ is the area of the source zone *s*.

### 2.3. Pollutant Emissions Data

In this paper, we used pollutant emissions data, rather than pollution concentrations data, to measure the environmental pollution impacts during urban expansion. Although pollution concentration data is an optional data source for this paper, it lacks the temporal coverage of our emission data. In addition, the pollution concentration data suffers from noise added by various spatial interpolation techniques and issues from selective choices of monitoring sites. The PM_2.5_ emission data was obtained from the School of Environmental Sciences at Peking University (http://inventory.pku.edu.cn), from which the monthly pollutant emissions data a spatial resolution of with 0.1 × 0.1 degrees may be freely downloaded.

To build the emission inventory, a space-for-time substitute method [35] was applied to calculate monthly (or daily) residential fuel consumption and, consequently, monthly emissions of individual pollutant emissions. The intra-annual variations associated with agricultural waste burning, deforestation, and wildfires were obtained directly from the Global Fire Emissions Database (GFED) [36]. Sub-national disaggregation of pollutant emissions was used to generate spatial resolution of 0.1 × 0.1 degrees [37]. The rural residential energy consumption data for China were updated based on a nationwide survey and a nationwide fuel weighing campaign conducted in 2012. The results are different from the International Energy Agency (IEA) and Food and Agriculture Organization of the United Nations (FAO) data, which overlook the rapid fuel mix transition in rural China.

The files with the extension “nc” contain monthly gridded emission data (1800 × 3600) provided in Network Common Data Form (NetCDF). Furthermore, we obtained pollutant emissions data that were processed from six sources, which are transportation (Tran), residential and commercial (RC), industry (Indu), energy production (EP), deforestation wildfire (DW), and agriculture (Agri) (Table 1). 

First, monthly gridded PM_2.5_ emission data in 1995, 2000, 2005, 2010, and 2014 for Beijing-Tianjin-Hebei region were downloaded according to the six categories above. Then we aggregated the monthly data to calculate the annual cumulative PM_2.5_ emissions of each county. Because the boundaries of counties in the study area and the grids of PM_2.5_ emission data do not match each other geographically, we used the standard GIS areal weighting approach for the data preprocessing. Finally, by dividing the annual cumulative PM_2.5_ emissions by the area of each county, we calculated the annual pollutant emission intensity of each county.

### 2.4. Bayesian Spatio-Temporal Dynamic Statistical Model

To explore the spatio-temporal dynamic characteristics between PM_2.5_ emissions and urbanization in each county of Beijing-Tianjin-Hebei region under the background of the long-term data series, we adopted a newly developed advanced Bayesian spatio-temporal statistical model, i.e., the Spatio-temporal Conditional Autoregressive (ST.CAR) model [38,39,40].

Specifically, we defined the 204 counties constituting the study area, which do not overlap each other, as Sn (*n* = 1, 2, …, 204) and the corresponding time periods as t. The Bayesian model we used here can be shown as:(2)$$y_{nt}|\mu_{nt}\;\sim \;f\left(y_{nt}|\mu_{nt},\;\mu_{nt}\right);\;n=1,2,\ldots ,204;t=1995,\;2000,\;\ldots ,\;2014$$
(3)$$\mu_{nt}=X_{nt}\sigma +\psi_{nt}$$
(4)$$\mu_{nt}=X_{nt}\theta +U_{nt}\eta +\psi_{nt}$$
where $y_{nt}$ means the observed urban expansion density (log LD) of the study area n at time t, following a normal distribution linked with variance σ^2^ and mean $\;\mu_{nt},\;N\;\left(\mu_{nt},\;\sigma^{2}\right)$; $U_{nt}$ means the urban expansion variables measured from the nighttime light data, that is LD of each county in the study area; $X_{nt}$ represents the selected six variables, log Tran, log EP, log Indu, log RC, log DW, and log Agri; θ and η represent the coefficient vectors to be estimated corresponding to each variable.

$\psi_{knt}$ represents a potential supplement for the study area n at the time period t, capturing the structured temporal and spatial random effects existing in the data collectively. To accurately model the spatial correlations, the N × N spatial weights matrix W was adopted to clarify the potential spatial connection structure among those counties (*n* = 1, 2,…, 204) [41,42]. 

The Bayesian spatio-temporal models we used in this paper were implemented by adopting the Bayesian Markov Chain Monte Carlo (MCMC) simulation approach, which is available in a free R software package named CARBayST [43]. For each of the models implemented, statistical inferences were based on two MCMC chains to ensure the convergence of samplers. In addition, the deviance information criterion (DIC) was applied to compare the simulation results of the Bayesian models [44].

### 2.5. Local Regression Model

To supplement the simulation results of the Bayesian spatio-temporal statistical model, and to further explore the potentially local changes existing in the data, a local regression model—a geographic weighted regression (GWR) model—was adopted in this study. In previous studies, the GWR model has been widely used to explore scientific issues associated with PM_2.5_ pollution [45,46,47]. The formula can be expressed as:(5)$$\ln y_{p}=\;\beta_{p0}+\overset{n}{\underset{p=1}{\sum }}\beta_{pk}x_{pk}+\;\varepsilon_{p}$$

In Equation (5), p(1,…, 204) gives the spatial location of each county; $y_{p}$ represents the log LD value of the p county (the dependent variable); six independent variables $x_{pk}\;\left(k=1,\ldots ,6\right)$, namely log Tran, log EP, log RC, log Indu, log DW, and log Agri; $\beta_{pk}$ represents the local regression parameters; and $\varepsilon_{p}$ represents the random error term, giving each county corresponding parameters to explore the correlation between urban expansion and PM_2.5_ emissions from six different sources.

## 3. Results

### 3.1. Characterization of the Spatio-Temporal Dynamics in Data 

Figure 2(1.a,1.b) reveals the spatio-temporal evolution of urban expansion and PM_2.5_ emissions in Beijing-Tianjin-Hebei region from 1995 to 2014. The classification method used in Figure 2, Figure 4, Figure 5 and Figure 6 was Natural Breaks (Jenks), meaning classes are divided based on the natural groupings inherent in data. The features are divided into multiple classes whose boundaries are set where the relatively large changes in the data exist. From 1995 to 2014, after two decades of development, the urban expansion in the Beijing-Tianjin-Hebei region made significant progress. Overall, the average value of log LD in the study area increased from 0.9397 in 1995 to 1.1392 in 2014. From the perspective of spatial changes, there were large differences in the speed of urban expansion in different regions. County-level administrative units located in the northwest of the Beijing-Tianjin-Hebei region, such as Kangbao County (the value of log LD was 0.38 in 1995 and 0.42 in 2014) and Guyuan County (the value of log LD was 0.41 in 1995 and 0.45 in 2014), maintained a low level of urban expansion and underwent little change during the past two decades; whereas county-level administrations located in the central and southeastern regions, such as Tongzhou (the value of log LD was 1.29 in 1995 and 1.68 in 2014) District and Yongnian County (the value of log LD was 0.94 in 1995 and 1.30 in 2014), experienced significant urban expansion. In addition, the urban agglomeration phenomenon in 2014 was more obvious than in 1995, particularly in Beijing, Tianjin, and surrounding cities.

From the perspective of PM_2.5_ emission intensity based on transportation, the average log Tran (4.4775) in Beijing-Tianjin-Hebei region in 2014 was significantly higher than that in 1994 (3.9147) as a whole (Figure 2(2.a,2.b)). Furthermore, in 1994, the county-level administrative units with high log Tran values were mainly concentrated in Beijing and Tianjin, whereas in Hebei Province, the counties’ values were relatively low and no obvious accumulation areas existed. After 20 years of development, regional differences in the study area increased significantly. PM_2.5_ emission intensities from transportation formed a series of agglomeration areas centered on Beijing, Tianjin, Shijiazhuang, Handan, and Tangshan.

Regarding PM_2.5_ emission intensity based on residential and commercial, the overall change in the counties in the Beijing-Tianjin-Hebei region during the past two decades has not changed significantly (average value of 6.2076 in 1995 and 6.4677 in 2014) (Figure 2(3.a,3.b)), and the spatial change trend is not obvious. Compared with 1995, the county-level administrative units with high log RC are still concentrated in the central and southern areas of the study area 20 years later.

Based on the average value alone, the PM_2.5_ emission intensity based on industry in the Beijing-Tianjin-Hebei region has not changed significantly overall (6.2749 in 1995 and 6.3509 in 2014) (Figure 2(4.a,4.b)). However, after two decades of development, the spatial distribution of these counties’ emission intensities changed significantly. The most obvious change is that the log Indu values of counties in Beijing and surrounding counties decreased. For example, the log Indu value of Mentougou District decreased from 6.84 in 1995 to 6.43 in 2014. However, the log Indu values of some county-level administrative units in Hebei Province increased significantly, such as Qianxi County (log Indu value of 6.20 in 1995 and 6.73 in 2014) and Gaocheng District (logIndu value of 6.67 in 1995 and 7.04 in 2014).

Similar to the PM_2.5_ emission intensity based on industry, the overall PM_2.5_ pollution caused by energy production in the Beijing-Tianjin-Hebei region did not change significantly (average value was 5.6675 in 1995 and 5.9665 in 2014) (Figure 2(5.a,5.b)). However, compared with 1994, the PM_2.5_ emission intensity based on energy production of each county in 2014 showed more spatially localized characteristics, of which the centralization phenomenon centered on Beijing, Tangshan, Handan, and Shijiazhuang was the most obvious.

In 2014, the county-level average of PM_2.5_ emission intensity from the deforestation wildfire sector in Beijing-Tianjin-Hebei region (3.8058) was significantly higher than that in 1994 (2.5092) (Figure 2(6.a,6.b)). The counties in the northwest of Beijing-Tianjin-Hebei region, such as Guyuan County (log DW value was 2.67 in 1995 and 4.66 in 2014), gradually became regions with high log DW values, whereas in 1994 the regions with high log DW values were concentrated in the central and southern regions, such as Ci County (the log DW value was 3.20 in 1995 and 4.01 in 2014).

Overall, the PM_2.5_ emission intensity from agriculture in the Beijing-Tianjin-Hebei region did not change significantly (average value of log Agri was 4.7293 in 1995 and 4.8050 in 2014) (Figure 2(7.a,7.b)). Regarding the change trend of its spatial distribution, the local accumulation was obvious in 1994, mainly concentrated in the Beijing-Tianjin area and Shijiazhuang and Handan areas in the south of Hebei Province; whereas two decades later, the counties in the east and south parts of the Beijing-Tianjin-Hebei area all became high log Agri value areas.

### 3.2. Results of Bayesian Spatio-Temporal Statistical Model Estimation

A set of Bayesian spatio-temporal statistical models were implemented to explore the dynamic spatio-temporal relationships between PM_2.5_ emission and urban expansion in Beijing-Tianjin-Hebei region from 1995 to 2014. 

First, we took the Beijing-Tianjin-Hebei region as a whole to explore the relationship between PM2.5 emission and urban expansion (model 1). We detected that the endogenous spatio-temporal autoregressive parameters of the data are quite high (both > 0.9) with narrow 95% confidence intervals, showing that the data has significant spatial correlation and time dependence, in addition to the necessity of spatio-temporal autoregressive statistical modeling.

Urban expansion in Beijing-Tianjin-Hebei Region was statistically significantly related to PM_2.5_ emission from transportation, industry, and energy production. Specifically, based on the estimated results (Table 2), with other conditions remaining the same, every 0.01 increase in log LD (roughly the average increase in log LD from 1995–2014) related to 7.89% (with a 95% confidence interval of [4.21%, 11.41%]) increase in log Tran; every 0.01 increase in log LD led to 13.6% (with a 95% confidence interval of [9.72%, 17.58%]) increase in log EP; and every 0.01 increase in log LD was associated with about 15.66% (with a 95% confidence interval of [10.62%, 20.41%]) increase in log Indu. By comparing the simulation results, the urban expansion of Beijing-Tianjin-Hebei area from 1995 to 2014 was most related to the PM_2.5_ emission intensity from industry, ceteris paribus.

Figure 3 shows the simulated values of urban expansion in Beijing-Tianjin-Hebei region over time based on model 1 from 1995 to 2014, and superimposes the upper and lower limits of these estimated values (shown by the dotted lines in the figure). The figure presents two interesting trends in the study area based on model simulation results. First, overall, the urban expansion of the counties in the Beijing-Tianjin-Hebei region has shown a gradual upward trend over time. Furthermore, in terms of growth rate, the growth rate was slower from 1995 to 2005, and a relatively high growth rate was maintained from 2005 to 2010; however, the growth rate slowed down again after 2010.

Figure 4 shows the estimated spatial distribution of urban expansion in Beijing-Tianjin-Hebei region from 1995 to 2014 after deducting the effect of covariates. The spatial correlation of these estimates clearly reveals the spatial agglomeration effect of the urban development intensity in the study area. They take Shijiazhuang, Handan, Tangshan, and Cangzhou of Beijing, Tianjin, Hebei Province as the central hot spots, and gradually reduce to their surrounding county-level administrative units.

Although the Beijing-Tianjin-Hebei integration policy has been implemented for many years (beginning in the 1980s), the socio-economic development levels of the three provinces of Beijing, Tianjin, and Hebei province remain uneven, and there are still significant gaps among these areas [48]. To further explore the relationship between urban expansion and PM_2.5_ emissions from different sources in the three provinces respectively, we established three new Bayesian spatio-temporal statistical models, model 2 (based on Beijing), model 3 (based on Tianjin), and model 4 (based on Hebei province). The simulation results are shown in Table 3.

Based on the simulation results of models 2, 3, and 4, the relationship between urban expansion and PM_2.5_ pollution in the three provinces is different. Urbanization was statistically significantly related to PM_2.5_ emission from transportation, and energy production in Beijing. Specifically, with other conditions remaining the same, every 0.01 increase in log LD led to 6.33% (with a 95% confidence interval of [0.57%, 12.68%]) increase in log Tran; every 0.01 increase in log LD was related to 42.66% (with a 95% confidence interval of [29.16%, 55.08%]) increase in log EP. Urbanization showed statistically significantly association with PM_2.5_ emission from industry in Tianjin. Every 0.01 increase in log LD led to 32.56% (with a 95% confidence interval of [17.12%, 48.43%]) increase in log Indu, ceteris paribus. Urbanization was statistically significantly related to PM_2.5_ emission from transportation, energy production, and industry in Hebei province. With other conditions remaining the same, every 0.01 increase in log LD (roughly the average increase in log LD from 1995–2014) led to 5.64% (with a 95% confidence interval of [1.12%, 10.53%]) increase in log Tran; every 0.01 increase in log LD was related to 11.79% (with a 95% confidence interval of [7.02%, 16.74%]) increase in log EP; and every 0.01 increase in log LD was related to 18.37% (with a 95% confidence interval of [12.40%, 24.66%]) increase in log Indu.

### 3.3. Results of Local Regression Model Estimation

As shown in Figure 3, the overall urban expansion in Beijing-Tianjin-Hebei region was the fastest in the time period from 2005 to 2010. Thus, to explore more specifically the trend of spatial correlation of PM_2.5_ emission and urban expansion over time in the context of rapid urban expansion, we adopted the geographic weighted regression (GWR) model, which is a local regression model. 

According to Table 4, the source of PM_2.5_ pollution in the Beijing-Tianjin-Hebei region most closely related to urban expansion in 2005 was industry (the coefficient of log Indu is 0.7108), followed by energy production (the coefficient of log EP is 0.7080), whereas the relationship between transportation and urban was relatively weak (the coefficient of log Tran is 0.5986). However, after five years of rapid development, the urban development pattern in the study area changed to a significant extent, and the correlation between the three major sources of PM_2.5_ pollution and urban expansion also changed significantly. Among these, the change of transportation is the most obvious, increasing from 0.5986 in 2005 to 0.7518 in 2010, whereas the relationship between urban expansion and the two other factors (industry and energy production) did not change significantly. Compared with those in 2005, the correlations between the two and urban expansion were even lower in 2010.

## 4. Discussion

Due to the acceleration of urbanization globally and the ensuing environmental pollution problems, clarifying the relationship between urban expansion and pollutant discharge is an important topic of theoretical and empirical study. Therefore, this paper adopted an advanced Bayesian spatio-temporal statistical model and a local regression model to measure the spatio-temporal dynamic relationships between PM_2.5_ emission sources and urban expansion.

Regarding the simulation results of the advanced Bayesian spatio-temporal statistical model, it can be inferred from the difference between the results of model 2, 3 and 4 that the industrial structures of Beijing, Tianjin, and Hebei provinces are quite different, and the PM_2.5_ emissions control policies need different emphases. The model also has limitations. This article mainly explores the relationship between different sources of PM_2.5_ emission and urban expansion, and therefore does not take into account some confounding variables that were associated with urbanization or PM_2.5_ emissions.

Regarding the simulation results of the GWR model, Figure 5 shows the spatial correlation between urban expansion and the PM_2.5_ emission from the transport sector in 2005 and 2010. Overall, the spatial correlation between urban expansion and the PM_2.5_ emission from transport sector in the Beijing-Tianjin-Hebei region increased, and we deduce that this is related to the popularity of family cars [49]. However, from 2005 to 2010, its spatial distribution did not change significantly, and both showed a gradual weakening trend from northwest to southeast, which is presumably related to the topographical factors of the study area (Figure 6) [50].

Some limitations remain. First, PM_2.5_ pollution is linked to both primary and secondary emissions. This paper mainly explores the correlation between PM_2.5_ pollution from different sources and urban expansion, so only primary PM_2.5_ emission is considered. In subsequent research, we will comprehensively analyze the correlation between urban expansion and PM_2.5_ from both primary and secondary emission. Second, with new credible data sources on urban expansion for other urban agglomerations in China, such as the Guangdong–Hong Kong–Macao Greater Bay Area, we will empirically check whether the correlation between PM_2.5_ emissions from different sources and urban expansion varies across different regions and discuss the potential mechanisms leading to such spatial heterogeneities. In addition, certain confounding variables that affect both pollutant emissions and urban expansion might be not incorporated in our model because of data limitations.

## 5. Conclusions

This paper explored the correlation between urban expansion and PM_2.5_ emissions from different sources in the Beijing-Tianjin-Hebei region. Our exploration mainly drew upon the fine resolution urban expansion indicator complied from remote sensing and satellite data sources. A Bayesian spatio-temporal autoregressive statistical model and a local regression model were used to explicitly analyze potential spatial correlations and temporal dependency between urban expansion and PM_2.5_ emissions from different sources. Our empirical results based on the Bayesian model showed that the urbanization in Beijing-Tianjin-Hebei region was significantly positively correlated with PM_2.5_ emissions from transportation, energy production, and industry, and model-based estimates of the spatial distributions of urban expansion presented obvious clustering patterns. Our empirical results based on the local regression model showed that the spatial correlations between PM_2.5_ emissions from different sources and urban expansion changed during the period of rapid urban expansion (2005–2010), and the biggest change originated from the PM_2.5_ emissions of the transport sector.

## Figures and Tables

**Figure 1 ijerph-18-00608-f001:**
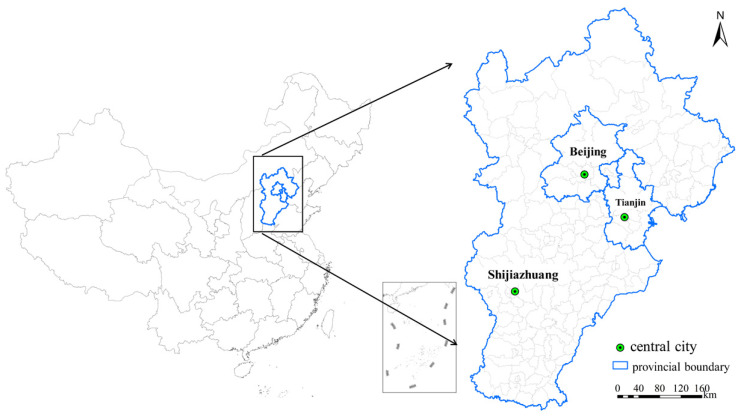
The study area.

**Figure 2 ijerph-18-00608-f002:**
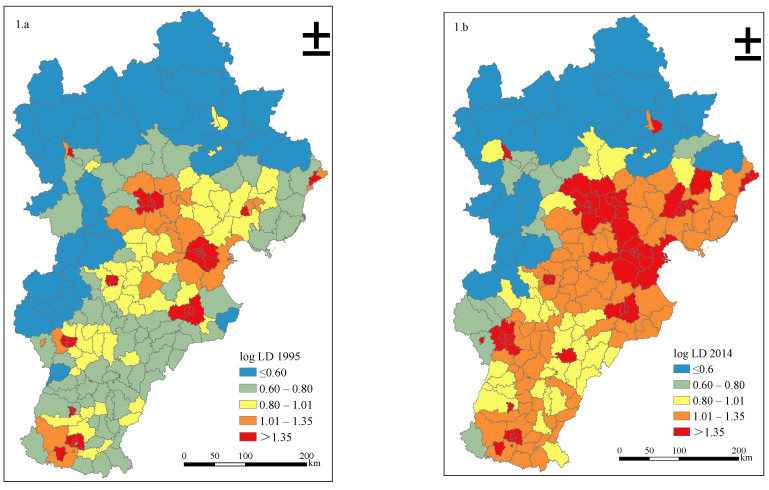
Spatio-temporal dynamics of log LD, log Tran, log RC, log Indu, log EP, log DW, and log Agri from 1995 to 2014 in Beijing-Tianjin-Hebei region.

**Figure 3 ijerph-18-00608-f003:**
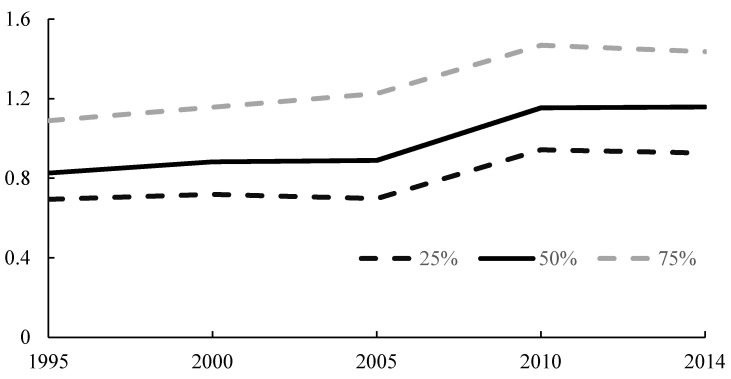
Model 1-based estimates of the temporal trends of urban expansion. The vertical axis legend represents the estimated log LD value of each county in the study area based on the model 1 simulation results. The three lines (75%, 50%, and 25%) represent the upper quantile, median, and lower quantile of estimation results on urban expansion after adjusting for covariate effects from 1994 to 2014.

**Figure 4 ijerph-18-00608-f004:**
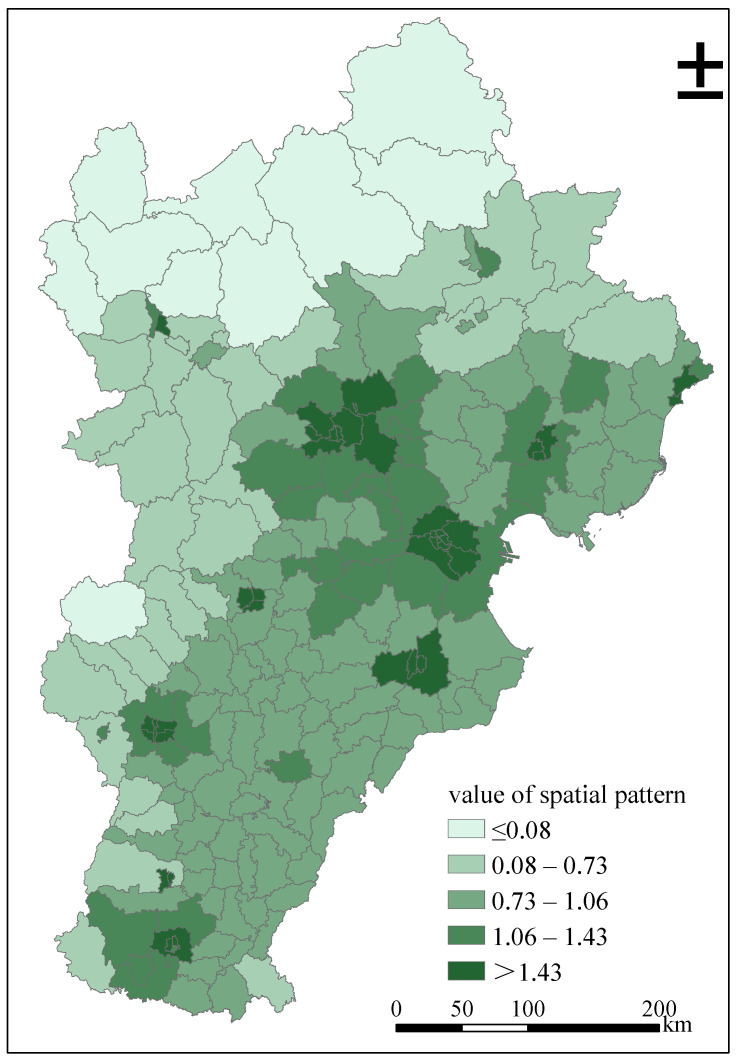
Estimated spatial patterns of urban expansion. Value of spatial pattern presents the estimates of spatial distributions of urban expansion, net of covariate effects.

**Figure 5 ijerph-18-00608-f005:**
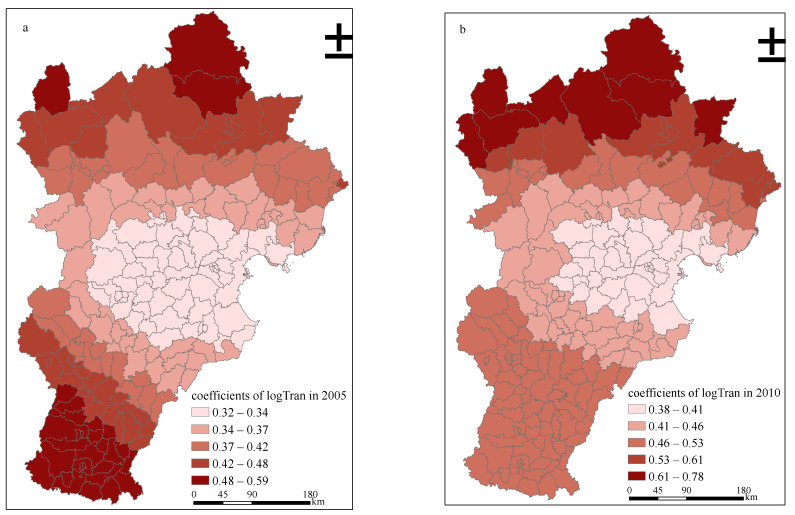
The results of log Tran based on the GWR model in Beijing-Tianjin-Hebei region in 2005 and 2010. (**a**) shows the degree of correlation between urban expansion and PM_2.5_ emission from the transport sector in the Beijing-Tianjin-Hebei region in 2005, while (**b**) shows that in 2010.

**Figure 6 ijerph-18-00608-f006:**
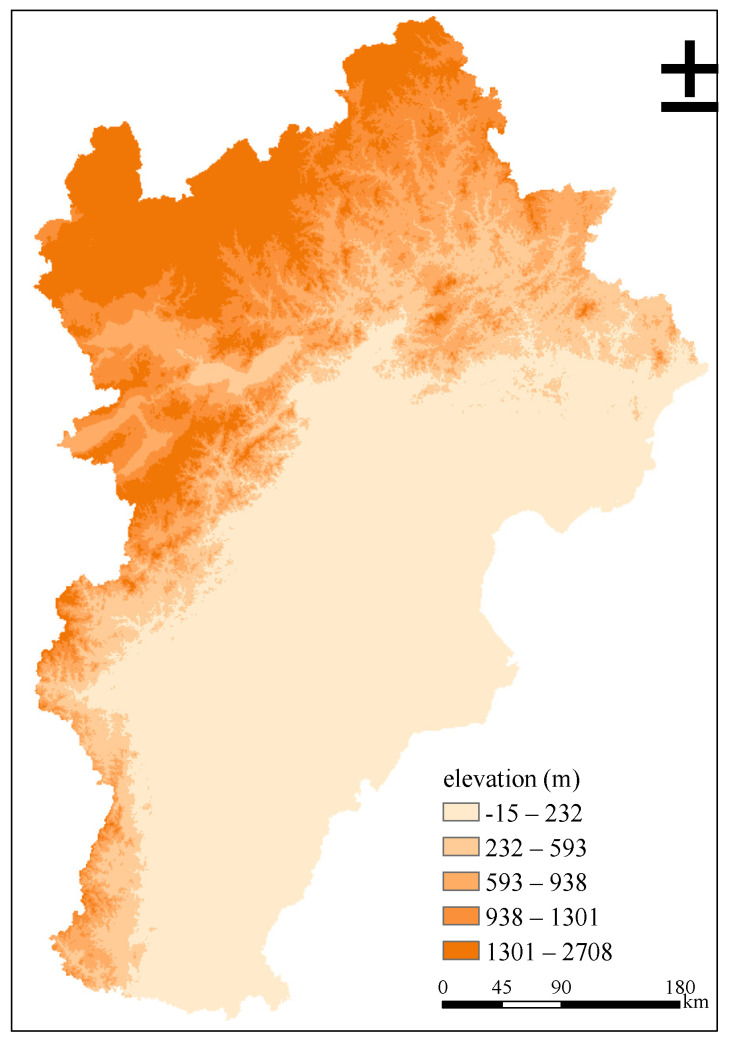
Elevation of Beijing-Tianjin-Hebei region.

**Table 1 ijerph-18-00608-t001:** Data and variables involved in this study.

Variables	Description	Mean	Standard Deviation
log LD	log of luminosity density	0.9397 (1995)	0.3726
		1.1392 (2014)	0.4632
log Tran	log of PM_2.5_ emission intensity from transport sector (g/km^2^)	3.9147 (1995)	0.6771
		4.4775 (2014)	0.6666
log RC	log of PM_2.5_ emission intensity from residential and commercial sector (g/km^2^)	6.2076(1995)	0.2712
		6.4677 (2014)	0.2736
log Indu	log of PM_2.5_ emission intensity from industry sector (g/km^2^)	6.2749 (1995)	0.5771
		6.3509 (2014)	0.5067
log EP	log of PM_2.5_ emission intensity from energy production sector (g/km^2^)	5.6675 (1995)	0.4506
		5.9665 (2014)	0.4501
log DW	log of PM_2.5_ emission intensity from deforestation wildfire sector (g/km^2^)	2.5092 (1995)	0.7176
		3.8058 (2014)	0.6937
log Agri	log of PM_2.5_ emission intensity from agriculture sector (g/km^2^)	4.7293 (1995)	0.4701
		4.8050 (2014)	0.3344

**Table 2 ijerph-18-00608-t002:** Model 1 estimation results.

Variables	Median	2.5%	97.5%
Intercept	−1.5643	−2.1456	−0.9747
log Tran	0.0789 *	0.0421	0.1141
log RC	0.0726	−0.0218	0.1629
log EP	0.1360 *	0.0972	0.1758
log Indu	0.1566 *	0.1062	0.2041
log DW	−0.0063	−0.0245	0.013
log Agri	0.0106	−0.0348	0.0541
*τ^2^*	0.0629	0.057	0.0694
*σ^2^*	0.0009	0.0006	0.0014
ρ	0.9598	0.9107	0.9874
*λ*	0.9138	0.8708	0.9561
DIC	−3348.4014		
Likelihood-value	2566.7320		

Note: “*” represents the statistical significance at confidence interval of 95%. To complete the specification of the Bayesian spatio-temporal autoregressive model, conventional prior distributions were specified for unknown model parameters: an inverse-gamma distribution for variance parameters ($\tau^{2}$ and $\;\sigma^{2}$); and a uniform distribution for spatial and temporal autoregressive parameters (ρ and *λ*).

**Table 3 ijerph-18-00608-t003:** Model 2, 3 and 4 estimation results.

Variables	Model 2			Model 3			Model 4		
	Median	2.5%	97.5%	Median	2.5%	97.5%	Median	2.5%	97.5%
Intercept	−3.1064	−4.1723	−2.0286	−2.1633	−3.5688	−0.7559	−1.2961	−2.0044	−0.6205
log Tran	0.0633 *	0.0057	0.1268	0.1129	−0.0024	0.2337	0.0564 *	0.0112	0.1053
log EP	0.4266 *	0.2916	0.5508	0.0608	−0.0379	0.1595	0.1179 *	0.0702	0.1674
log Indu	0.0431	−0.1232	0.2117	0.3256 *	0.1712	0.4843	0.1837 *	0.1240	0.2466
*τ^2^*	0.0160	0.0084	0.0292	0.0159	0.0103	0.0244	0.0601	0.0538	0.0671
*σ^2^*	0.0024	0.0012	0.0049	0.0017	0.0009	0.0033	0.0010	0.0007	0.0015
ρ	0.2858	0.0307	0.7181	0.7388	0.4229	0.9317	0.9209	0.8643	0.9626
*λ*	0.9165	0.7604	0.9952	0.7384	0.4951	0.9469	0.9163	0.8664	0.9616
DIC	−207.5416			−240.9145			−2754.7026		
Likelihood-value	157.8441			174.3942			2124.7144		

Note: “*” represents the statistical significance at confidence interval of 95%. The estimation results of model 2 based on Beijing; the estimation results of model 3 based on Tianjin; the estimation results of model 4 based on Hebei.

**Table 4 ijerph-18-00608-t004:** Results of the correlation analysis between urban expansion and pollutant emissions based on the geographic weighted regression (GWR) model (N = 204).

Variables	2005	2010
log Tran	0.5986 *	0.7518 *
log EP	0.7080 *	0.6676 *
log Indu	0.7108 *	0.7062 *

Note: “*” represents the statistical significance at confidence interval of 95%.

## Data Availability

The data belongs to Institute of Geographic Sciences and Natural Resources Research, Chinese Academy of Sciences, and the data can be available on request, but the intention of using the data must be for the purpose of conducting research, then the disclosure include the following: The title of the research or paper for which the specified data is to be used; The details of the institution and supervisory body or persons under the auspices of the research is undertaken; The assurance that no commercial gain will be received from the outcome from the research.

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
