# Peer review of "Exploring the Dynamic Spatio-Temporal Correlations between PM2.5 Emissions from Different Sources and Urban Expansion in Beijing-Tianjin-Hebei Region"

_ijerph, 2021, doi:10.3390/ijerph18020608_

Round 1
Reviewer 1 Report
See pdf attached
Note Same color scales are required when maps are compared.

Author Response
Dear reviewer,
Thank you very much for your generous help to improve our manuscript entitled “Exploring the Dynamic Spatio-temporal Correlations between PM2.5 Emissions from Different Sources and Urban Expansion in Beijing-Tianjin-Hebei Region”. We appreciate all of your comments and hope that the revised manuscript will be eligible for publication. The responses to your valuable suggestions are attached as a PDF file.

Reviewer 2 Report
Dear Authors,
The present work is of interest to the scientific community, society and the environment. It is known that the emission of PM2.5 causes several damages to the health of the population directly affected. The focus of the work is unprecedented, when referring to statistical modeling and approach in the analysis of the temporal dynamics x spatial corrections of MP2.5.
However, I did not identify in the study an approach to the treatment of meteorological data from the emitting sources, such as wind roses. Although the study area is very extensive, these meteorological parameters would further corroborate your modeling.
Author Response

(The authors gave the same response as above.)

Reviewer 3 Report
Overall the article is well structured and written. However, there are some minor issues regarding the spelling. Some specific comments are
-Check the years used in the text. In line 119, 2020 has been written mistakenly in place of 2010. Check other years as well.
-Check spaces used between words. I some places, there are more than one space.
Author Response

(The authors gave the same response as above.)

Reviewer 4 Report
Comments to Author:
I have found the object of the paper interesting and appropriate for the journal. The paper is well-organized however, there are some issues that I would like you work on, in order (I hope) to increase the quality of work. In what follow, my comments:
L15: “statistical model modeling”. Please, use a synonymous for model or for modeling.
L81-82-83: Pease, replace “chapter” with “section”.
L108: ”expansion.Here” add a space
L108: I think it is better to write “luminosity density (LD) to” instead of “LD (luminosity density) to”
L108-109: Please, add a explanation at least as footnote for the “The standard GIS areal
109 weighting approach” or add a reference.
L111: Please, better motivate the choice of data on pollutant emissions instead of data on pollution concentrations.
L132: Since N=204, I think it is better to write “n= 1, 2, …, 204” instead of “n= 1, 2, …, N”.
L132: Please, specify the periods t=1,..,5
L134-135: I think it is better to write “at time t” or “at period t” instead of “at time period t”
L135: There is a typo in the formula of variance.
L142: “by adopted” I think it is better to write “by adopting”
L142-145: I think that Authors should add additional information about the model for those are not familiar with it.
L154: There is a missing space “county(the dependent variable) “
L221: Since the figures are splitted on several pages, I think it is better to enumerate it separately adding a description for each figure. Moreover, I think it better to incorporate the following text in the paper since it refers to several figures. “The classification method used in Figure 2, Figure 4, Figure 5, and Figure 6 was Natural Breaks (Jenks), meaning classes are divided based on the natural groupings inherent in data. The features are divided into multiple classes whose boundaries are set where the relatively big jumps in the data are.”
L250: The description of Figure 3 should be placed on the same page as the figure.
L255: I think it is better to use a color scale (as in Figure 2).
L315: Please, check the expression “…significance at 95% credible interval of 95%. “
L336: I think it is better to use a color scale (as in Figure 2).
L339: I think it is better to use a color scale (as in Figure 2).
OVERALL: Please check for English.
Author Response

(The authors gave the same response as above.)

Reviewer 5 Report
The goal of this study was evaluating the relationship between PM2.5 emission and urban expansion. I think that presented proposal is worth attention and the issues in the paper are current. The authors have very well identified the research problem.
However, a major drawback of this article is unclear research hypothesis.
As authors mentioned in line 71~79, this study was conducted to provide reliable and specific 72 estimates of the correlation between PM2.5 emission and urban expansion using emission data (20 years-validated in previous paper), remote Sensing data (night time light data) to characterize urban expansion and bayesian spatio-temporal autoregressive model and local regression model
Then, authors chose night time light data (log LD) as dependent (Y, so called response variable) variable and PM emission from various sources as independent (X, so called explanatory variable) variable in the Bayesian model as well as local regression model.
It looks that using the variable of the light data, representing urban expansion, as explanatory variable while PM emission, as response variable, is much more interpretable and understandable.
Change the response variable or provide much clear rationale why you choose PM emission as independent variable and light data as dependent variable in your current model. The author needs to clarify what is the study goal (the impact of expansion of urbanization on PM emission? or the impact of PM emission on expansion of urbanization? with clear explanation of corresponding model variables) in abstract and main body.
And limitation of this study is also missing.
Conclusion is highly unacceptable and it does not focus on the empirical finding and the stated research goal. Also conclusion should be much concise and shorten.
Implication for future may also be included in the conclusion at the end.
Minor,
- Currently, the number of word for abstract is 267, Please check the limit of word count for Abstract and if possible, reduce them to fit to the limit.
- Line 80~84 : provide sentences with past tense
- provide the meaning of abbreviations (LD, Tran, RC, Indu, EP, DW, Agri) below the table 1
- Include discussion about how much the results will be different if authors use pollution concentration data, rather than, pollution emission data.
Author Response

(The authors gave the same response as above.)
